# Enhancement of enzymatic activity by biomolecular condensates through pH buffering

F. Stoffel[1], M. Papp[1], M. Gil-Garcia [1], A. M. Küffner[1], A. I. Benítez-Mateos [1], R. P. B. Jacquat [1], N. Galvanetto [2], L. Faltova[1] & P. Arosio [1] ✉

Biomolecular condensates can affect enzymatic reactions by locally changing not only concentrations of molecules but also their environment. Since protein conformations can differ between the dense and dilute phase, phase separation can particularly modulate enzymes characterized by a conformation-dependent activity. Here, we generate enzymatic condensates containing a lipase from *Bacillus thermocatenulatus*, which exhibits an equilibrium between a closed, inactive state, and an open, active conformation. We show that the activity of the enzyme increases inside the dense phase, leading to an enhancement of the overall reaction rate in the phase-separated system. Moreover, we demonstrate that these condensates can generate a more basic environment compared to the surrounding solution, maintaining a high enzymatic activity even in a solution pH interval that would be otherwise less favorable for the lipase. We further show that the formation of two phases with distinct pH values optimizes a cascade reaction involving two enzymes with different optimal pH conditions. Our results demonstrate that, through local pH buffering, biomolecular condensates can expand the optimal pH interval for enzymatic reactions and increase their robustness towards changes in environmental parameters. These findings have implications in biology and biotechnology for biocatalytic engineering, for instance for enabling network reactions with enzymes that require distinct pH values.

Evidence suggests that cells can regulate biochemical activity spatially and temporally through membraneless organelles, which are also referred to as biomolecular condensates[1–3]. Various membraneless organelles have been associated with different cellular functions, including enzymatic reactions[4]. Despite the increasing number of observations correlating enzymatic reactions with membraneless organelles[5–12], the molecular mechanisms underlying the potential role of biomolecular condensates in modulating enzymatic reactions remain largely unclear. Unraveling these mechanisms is crucial not only for fundamental biology but also for biotechnological applications, since condensates could represent a novel attractive technology to modulate enzymatic activity[13–22].

Biomolecular condensates form via phase separation of proteins often in combination with nucleic acids, resulting in the formation of a high-concentration phase surrounded by a lower-concentration phase. A first effect of biomolecular condensates on enzymatic reactions is therefore a local increase in protein concentration, which can lead to a local increase in reaction rates. Moreover, these condensates can function as molecular sorters, controlling the partitioning of different molecules in the two phases[23]. This effect is particularly relevant if cofactors or inhibitors of the reaction are excluded or strongly partitioned in the dense phase.

Alongside variations in local concentrations, emerging evidence suggests that biomolecular condensates can exert other significant

[1]Institute for Chemical and Bioengineering, Department of Chemistry and Applied Biosciences, ETH Zurich, Zurich, Switzerland. [2]Department of Biochemistry, University of Zurich, Zurich, Switzerland. ✉e-mail: paolo.arosio@chem.ethz.ch

effects. These include the formation of an interface between the dense and dilute phase[24–29], and changes of the local environment within the dense phase compared to the dilute phase[13,15,18,23,30,31], such as water thermodynamics[32]. Condensates can therefore modulate reactions by acting as an effective solvent with distinct physico-chemical properties compared to the surrounding phase[15,18,19,30,33–35]. This effect could be equally or even more important than local variations in concentration since changes in the environment can affect enzymatic activity.

We therefore expect that the effect of condensation can be particularly relevant for enzymes that are prone to conformational changes. Prompted by this hypothesis, in this work we designed condensates based on the enzyme *Bacillus thermocatenulatus* Lipase 2 (BTL2), which is prone to conformational changes. This enzyme is used in a wide array of biotechnological applications across the food, medicinal, and biofuel industries[36]. Like many members of the lipase family[37], BTL2 can transition from a closed, inactive state to an open, active state through extensive rearrangement of the lid domain, which results in the exposure of a hydrophobic binding pocket (Fig. 1a)[38].

It has been shown that the active, open conformation can be promoted by at least two mechanisms. Firstly, the multiple salt bridges that stabilize the open conformation can be favored in more hydrophobic environments than water (e.g., an alcohol-water mixture)[39–41]. Secondly, it has been recently demonstrated that a lipase from *Thermomyces lanuginosus* exhibits a higher population of the enzyme in the open conformation at increased protein concentrations due to enhanced oligomerization[42]. Since condensates have been shown to be more apolar than water[13,43] and locally increase enzyme concentration up to thousands-fold, we postulated that these two combined effects will increase the activity of BTL2 within condensates by stabilizing the open conformation.

Here, we demonstrate that indeed biomolecular condensates increase the enzymatic rate of BTL2. Moreover, we show how this local increase in enzymatic reaction rate, together with local alteration in pH inside the condensates, can significantly alter the dependence of the enzymatic rate on changes in the solution pH. These findings highlight a possible function of biomolecular condensates, which has crucial implications in both fundamental research and bioengineering. Indeed, this strategy can expand the optimal pH range for enzymatic activity and increase the robustness of enzymatic reactions upon changes in environmental parameters. Moreover, here we show that the pH buffering induced by condensates can enable enzymatic cascade reactions that would be otherwise incompatible in solution.

## Results

### Condensation increases lipase enzymatic activity

Following a strategy previously established in our group, we conjugated the BTL2 enzyme at both the N- and the C-terminus with the RGG intrinsically disordered region of the DEAD-box protein Laf1, which drives phase separation of the entire chimeric construct (Fig. 1b)[13–15]. The full sequence of this chimeric construct, indicated in the following as Laf1-BTL2-Laf1, is reported in Supplementary Fig. 1. We first verified the formation of Laf1-BTL2-Laf1 condensates by bright field and fluorescence confocal microscopy in a broad range of solution conditions, including the reference 24 mM Tris buffer, 10 mM NaCl at pH 7.5 (Fig. 1c, d and Supplementary Fig. 2).

We next evaluated the amount of enzyme recruited in the dense phase by removing the condensates by centrifugation and measuring the amount of enzyme in the supernatant by size exclusion chromatography (SEC). The results revealed that 93% of the Laf1-BTL2-Laf1 protein is recruited into the condensates (Table 1 and Supplementary Fig. 3). The total volume fraction ($\phi$) of the dense phase was evaluated by conducting a z-stack analysis with confocal microscopy (see Materials and Methods), obtaining a value of approximately 0.017%. Given the mass balance equation $\phi = \frac{c_{tot} - c_{dil}}{c_{dense} - c_{dil}}$, this value corresponds to a concentration in the dense phase of approximately 2.7 mM. This

indicates a high enzyme partitioning $K_E = \frac{c_{dense}}{c_{dil}}$ of 73,000. These values, summarized in Table 1, are overall consistent with the results obtained previously with similar chimeric constructs[13,15].

To show that the environment of the condensates is indeed less polar than water, we measured the spectrum of the environmentally sensitive dye PRODAN, which is influenced by the polarity of its surrounding environment[13,44,45]. The measured fluorescence emission maximum ($\lambda_{max}$) of this fluorophore within the Laf1-BTL2-Laf1 condensates is comparable to the value measured in isopropanol (Fig. 1e), confirming that the environment of the condensates is less polar than water.

We therefore next assessed whether the environment of biomolecular condensates can increase enzymatic reaction similarly to organic solvents. To this end, as a reference, we first measured the enzymatic rate of BTL2 in Tris buffer in the presence and absence of 10% isopropanol (Fig. 1f). As a model reaction we selected the hydrolysis of the analytical substrate 4-Methyl Umbelliferone Butyrate (MUB) into the fluorescent product 4-Methyl Umbelliferone (MU)[39,46] (Fig. 1f). The progress of the reaction can be therefore monitored by recording the increase of the fluorescent signal of the product over time. From the full reaction profile, we extracted the initial rates, as described in Materials and Methods (see also Supplementary Fig. 4). The results revealed an approximately 6-fold increase in the reaction rate in the presence of 10% isopropanol (Fig. 1f).

We verified that the dense phase does not interfere with the fluorescence signal of MUB by measuring the fluorescence of the molecule in the absence and presence of Laf1-BTL2-Laf1 condensates. No significant difference between the two conditions was observed (Supplementary Fig. 5). Therefore, the same measurement was then repeated for the condensate system. Representative kinetic profiles for the BTL2 and Laf1-BTL2-Laf1 systems at the same total concentrations of 0.5 μM enzyme and 250 μM substrate are shown in Fig. 1g. We observed a 3-fold increase in the overall initial rate in the presence of condensates (Fig. 1g, h), which is comparable in order of magnitude to the increase observed with the addition of 10% isopropanol. We note that for the heterogeneous system, the extracted initial rate represents the average contribution of both the dilute and the dense phase.

To demonstrate that this effect arises from the formation of biomolecular condensates and not from the conjugation of the enzyme with the intrinsically disordered region, we repeated the experiments using a buffer containing 750 mM NaCl. At this condition, Laf1-BTL2-Laf1 condensates are dissolved (Fig. 1d and Supplementary Fig. 2). We observed no rate enhancement in this condition (Fig. 1h), thus confirming that the observed increase in the initial rate 10 mM NaCl is indeed due to condensation. To further support this finding, we measured the enzymatic rate at 10 mM NaCl and low enzyme concentration (20 nM), where no condensates form (see also Supplementary Fig. 3). Again, no difference in reaction rate was observed between the BTL2 and Laf1-BTL2-Laf1 systems (Fig. 1h). This important result indicates that in this system, the activity of the enzyme in the dilute phase of the heterogeneous system is comparable to that of the homogeneous BTL2 solution.

Another important effect of condensation is the formation of an interface between the dense and dilute phases. Recent works have shown that the different concentrations of ions in the dense and dilute phases establish an electric potential difference across the interface of condensates[26–28]. Such potential could induce an inherent catalytic ability of condensates for a hydrolysis reaction as the one considered in this work[26]. We performed control experiments with condensates composed only of the RGG domain of Laf1 without the BTL2 enzyme. We did not observe any hydrolysis reaction of the MUB substrate in this system (Supplementary Fig. 6), thereby confirming that in our system, the catalytic activity is due to the BTL2 enzyme.

To further analyze the observed rate enhancement in the heterogenous condensate system, we performed experiments at

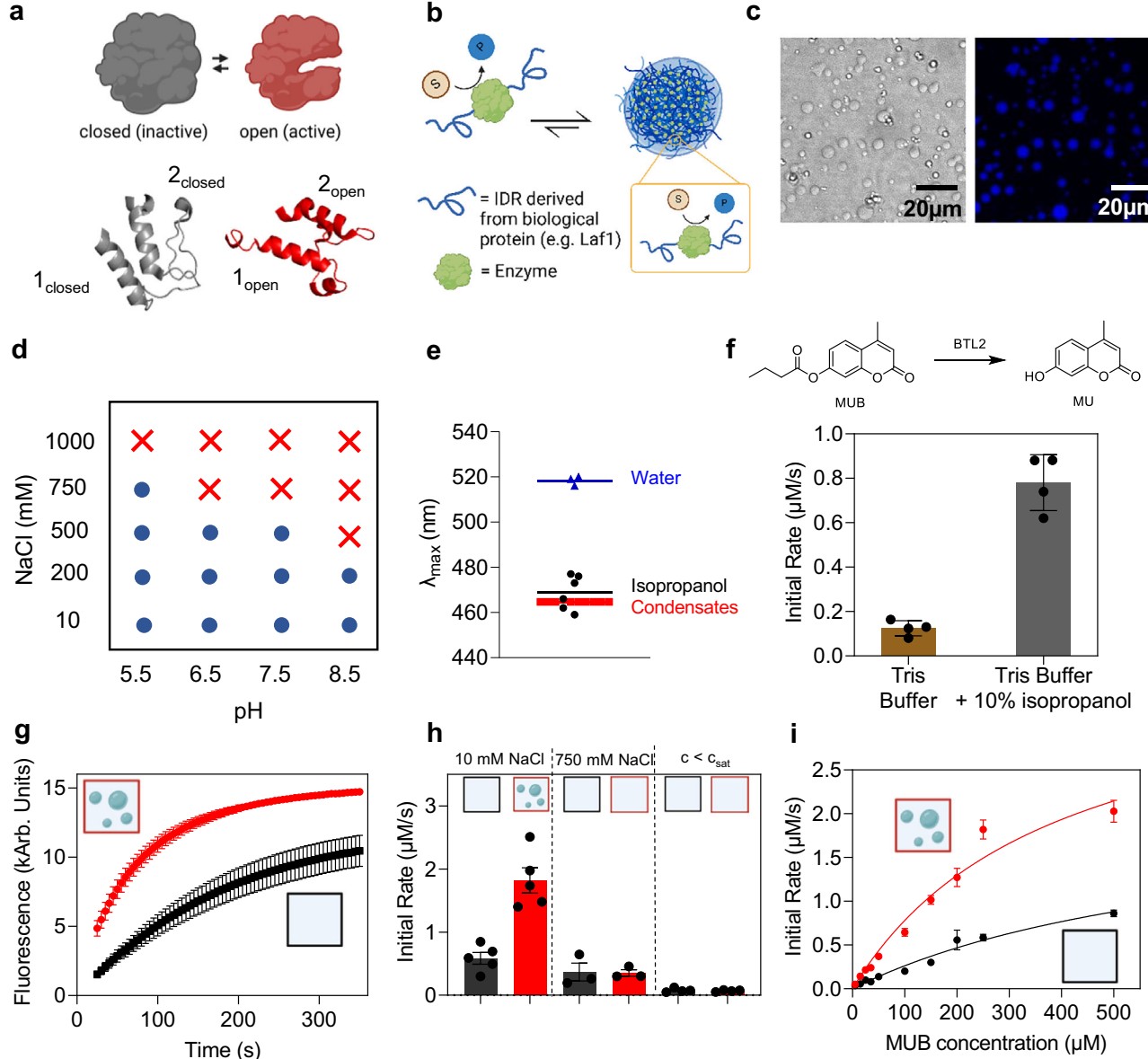

**Fig. 1 | Microreactor design and characterization. a** Schematic illustration of the conformational change of BTL2 (red and gray regions denote the open and closed conformations, respectively). PDB open: 2w22, PDB closed: 1ji3 (analog from *G. Stearothermophilus*). Secondary structures are visualized with PyMOL™ 2.5.4 **b** Schematic representation of the design of the fusion protein forming condensates. **c** Brightfield and fluorescence confocal images of Laf1-BTL2-Laf1 condensates stained with 25 μM SNARF-1 (0.5 μM protein, 24 mM Tris Buffer at pH 7.5, 10 mM NaCl). Fluorescence intensity was maximized for visualization purposes. **d** Phase diagram of Laf1-BTL2-Laf1 measured at 0.5 μM protein concentration at different pH and salt conditions. Blue circles and red crosses denote the presence and absence of phase separation, respectively. All buffers contain Tris/ BisTris Buffers with 10 mM ionic strength **e** PRODAN assay to evaluate the apparent polarity of the Laf1-BTL2-Laf1 condensate environment (9 replicates). Fluorescence emission maximum $\lambda_{max}$ measured at 0.5 μM Laf1-BTL2-Laf1 in 24 mM Tris at pH 7.5 and 10 mM NaCl, 100 μM PRODAN. For comparison, values measured in water (3 replicates) and isopropanol (6 replicates) are included. **f** Initial rates of MUB hydrolysis catalyzed by BTL2 in the absence and presence of 10% isopropanol (0.1 mM BTL2, 0.25 mM MUB, 24 mM Tris, pH 7.5, 4 replicates each). Error bars

represent the standard error of the mean. **g** Representative kinetic curves of the MUB hydrolysis reaction catalyzed by BTL2 (black) and Laf1-BTL2-Laf1 (red) (0.5 μM protein, 0.25 mM MUB, 10 mM NaCl, 24 mM Tris, pH 7.5, 5 replicates each). Error bars denote the standard error of the mean. **h** Initial rates of MUB hydrolysis by BTL2 (black) and Laf1-BTL2-Laf1 (red) in 24 mM Tris, pH 7.5 at low ionic strength (10 mM NaCl, and 0.5 μM enzyme, 5 replicates), high ionic strength (750 mM NaCl and 0.5 μM enzyme, 3 replicates) and at an enzyme concentration below $c_{sat}$ (10 mM NaCl and 0.02 μM enzyme, 4 replicates). All experiments were performed with 0.25 mM MUB. Error bars represent the standard error of the mean. **i** Average enzymatic rate of BTL2 (black) and Laf1-BTL2-Laf1 (red) measured at increasing substrate concentrations (0.5 μM enzyme, 24 mM Tris, 10 mM NaCl, pH 7.5, at least 4 replicates). Continuous lines represent fits according to a Michaelis-Menten model. Error bars represent the standard error of the estimate. Source data for panels **e–i** are provided as a Source Data file. Panels **g–i**: created in BioRender. Stoffel, F. (2025) https://BioRender.com/sfpppex, https://BioRender.com/9ecyox6; panel **a**: created in BioRender. Stoffel, F. (2025) https://BioRender.com/bb8rbcq; panel **b**: created in BioRender. Stoffel, F. (2025) https://BioRender.com/r5e1O5g.

increasing substrate concentrations for both the homogeneous BTL2 and heterogeneous Laf1-BTL2-Laf1 systems (Fig. 1i and Supplementary Fig. 7). It should be noted that substrate solubility did not allow to measure reactions at higher substrate concentration closer to the $K_M$.

The measured initial rates can be well captured by a simple Michaelis-Menten model, which showed a two-fold increase in apparent $v_{max}$ and a halving of the apparent $K_M$ in the heterogeneous system compared to homogeneous BTL2 (Fig. 1i and Table 2). Again, we note that for the

## Table 1 | Characterization of Laf1-BTL2-Laf1 condensates

| Volume Fraction ($\phi$) | % Protein in Dense Phase | $c_{dil}$ ($\mu M$) | $c_{dense}$ (mM) |
|---|---|---|---|
| 0.017 | 93 | 0.036 | 2.7 |

Total volume fraction of the dense phase, fraction of enzyme recruited into the dense phase, and concentrations of enzyme in the dense and dilute phases in 24 mM Tris buffer at pH 7.5 with 10 mM NaCl.

## Table 2 | Kinetic constants of homogenous and heterogenous lipase systems

|  | Homogenous | Heterogenous |
|---|---|---|
| $K_M$ ($\mu M$) | $713 \pm 281$ | $388 \pm 128$ |
| $v_{max}$ ($\mu Ms^{-1}$) | $2.1 \pm 0.6$ | $3.8 \pm 0.7$ |
| $k_{cat}$ ($s^{-1}$) | $4.3 \pm 1.2$ | $7.6 \pm 1.4$ |

Laf1-BTL2-Laf1 heterogeneous system, these values represent only apparent constants that average the contribution of the dilute and dense phase.

### Kinetic model analysis

To quantify the individual contributions of the dilute and dense phases, we analyzed the experimental data with a kinetic model that considers enzymatic reaction occurring in both phases according to Michaelis–Menten kinetics in the absence of mass transfer limitations. The detailed derivation of the kinetic model can be found in the Supplementary Information. Based on this model, the ratio between the initial rates measured in the heterogeneous and homogeneous systems at the same total protein concentration, indicated as $\frac{r_{het}}{r_{hom}}$, can be expressed as:

$$\frac{r_{het}}{r_{hom}} = \xi \cdot (1 - \Phi_D) + \xi \cdot \Phi_D \cdot \frac{k_{cat}^{II}}{k_{cat}^{I}} \cdot K_E \cdot \frac{K_M^{*I} + [S]}{\frac{K_M^{*II}}{\gamma_S^{II} \cdot K_S} + [S]} \quad (1)$$

where the upper indices $I$ and $II$ denote the dilute and dense phase, respectively, $\Phi_D = \frac{V^{II}}{V^{I} + V^{II}}$ is the volume fraction of the dense phase, $\xi = \frac{[E]_0^I}{[E]_0^{hom}} = \frac{1}{K_E \cdot \Phi_D + 1 - \Phi_D}$ is the ratio of enzyme concentrations in the dilute phase in the heterogeneous and homogeneous systems at the same protein concentrations. $K_E = \frac{[E]_0^{II}}{[E]_0^{I}}$ and $K_S = \frac{[S]^{II}}{[S]^{I}}$ are the enzyme and substrate partition coefficients respectively and $\gamma_S$ and $[S]$ denote the activity coefficient and the concentration of the substrate respectively. The values for $k_{cat}^{I}, K_M^{*I}, K_E, K_S$ and $\Phi_D$ were measured experimentally (see Supplementary Information).

Importantly, in our enzymatic condensates, the intermolecular interactions leading to phase separation are mainly driven by the intrinsically disordered domains of the chimeric protein. Hence, the activity coefficient of the enzyme is the same in the dilute and dense phase. The activity coefficient of the substrate in the dilute phase was assumed to be equal to one. Moreover, based on the results discussed previously (Fig. 1h), the kinetic constants $k_{cat}^{I}$ and $K_M^{*I}$ were assumed equal to the homogeneous system.

The kinetic constants in the dense phase $k_{cat}^{II}$ and $\frac{K_M^{*II}}{\gamma_S^{II} \cdot K_S}$ were fitted from the $\frac{r_{het}}{r_{hom}}$ values using Eq. 1, where $r_{het}$ denotes experimental initial reaction rates in the heterogeneous system and $r_{hom}$ corresponds to the fitted Michaelis-Menten rates for the homogeneous system. $r_{het}$ was estimated considering data from two biological replicates. Following this strategy, the $k_{cat}$ in the dense phase was estimated to be equal to $6.9 \, s^{-1}$.

The evaluation of the apparent Michaelis-Menten constant $K_M^{*II}$ requires an assumption about the activity coefficient of the substrate in the dense phase, which is challenging to measure experimentally.

Our model is built on the assumptions of instantaneous phase equilibrium, in which case the activity coefficient of the substrate in the dense phase can be assumed equal to $\gamma_S^{II} = \frac{1}{K_S}$[47], leading to $K_M^{*II} = 334 \, \mu M$. This reflects a two-fold higher affinity of the substrate in the dense phase compared to homogeneous solution. Moreover, we note that our model does not consider diffusion limitations. However, using fluorescence correlation spectroscopy (FCS) we determined the diffusion coefficient of a small molecule (Atto565) to be $1.015 \, \mu ms^{-1}$, which, considering a condensate diameter of $1 \, \mu m$, corresponds to a characteristic time of diffusion ($\tau_{diff}$) of 985 ms (see Materials and Methods and Supplementary Fig. 8). This is in the same order of magnitude as the characteristic time of reaction ($t_{react} = 1/k_{cat} = 145 \, ms$). Therefore, the fitted kinetic parameters for the dense phase with the simplified model may even underestimate the increase in $k_{cat}$ and decrease in $K_M$.

Our kinetic model allows to compare the amount of product formed in the two phases according to Eq. (2) (see derivation in Supporting equation (15))(Supplementary Fig. 9):

$$\frac{dn_{dense}}{dn_{dilute}} = \frac{\Phi_D}{1 - \Phi_D} \cdot \frac{k_{cat}^{II}}{k_{cat}^{I}} \cdot K_E \cdot \frac{K_M^{*I} + [S]}{\frac{K_M^{*II}}{\gamma_S^{II} \cdot K_S} + [S]} \quad (2)$$

We estimated that at all substrate concentrations, most of the product (approximately 95%) is formed inside the condensates (Supplementary Fig. 9). This result, together with the good agreement between the kinetic parameters obtained with a simple one-phase Michaelis-Menten model (Fig. 1i, Table 2) and the more complex kinetic scheme considering instantaneous phase equilibrium[47] in the heterogeneous system (Fig. 2b), support our assumptions regarding the activity coefficient of the substrate in the dense phase.

Overall, the kinetic analysis confirms that the higher overall initial rate observed for the heterogeneous system is due to a change in the enzyme's kinetic parameters within the condensates. It is very likely that this increase in enzymatic activity is due to a conformational change of the enzyme within the condensates (Fig. 1a), driven by the more apolar environment and the local increase in enzyme concentration.

### Condensates can change the dependence of enzymatic reactions on solution pH

In addition to affecting the effective polarity, recent studies have shown that protonation states can be altered within condensates, leading to possible local changes in pH[16,26,27,48–52]. Since lipase activity is modulated by both the protonation state of the catalytic serine residue and the nucleophilic water molecule involved in the hydrolysis reaction[53], it is important to evaluate the effect of the possible change of the protonation state inside the condensates. We measured variations in the protonation state of the SNARF-1 dye via fluorescence confocal microscopy (see Materials and Methods). The spectrum of this small molecule is characterized by two emission maxima (580 nm and 640 nm) which change in relative intensity as a function of the protonation state of the molecule (Supplementary Fig. 10). Using a calibration curve (Supplementary Fig. 10), we observed an increase in the local apparent pH from 7.5 to 8.0. Control experiments measuring the enzymatic activity of BTL2 at pH 8.0 showed no increase compared to pH 7.5 (Supplementary Fig. 10).

However, the change of apparent pH inside the dense phase opens up the attractive possibility to explore whether condensation could affect the dependence of the enzymatic rate on changes in the solution pH by locally changing the protonation state of enzyme and clients inside condensates.

We performed these experiments with a second chimeric construct, which has the same architecture as Laf1-BTL2-Laf1 but is based on the intrinsically disordered domain of the DEAD-box protein DDX4[30]. The full sequence of this construct, indicated in the following

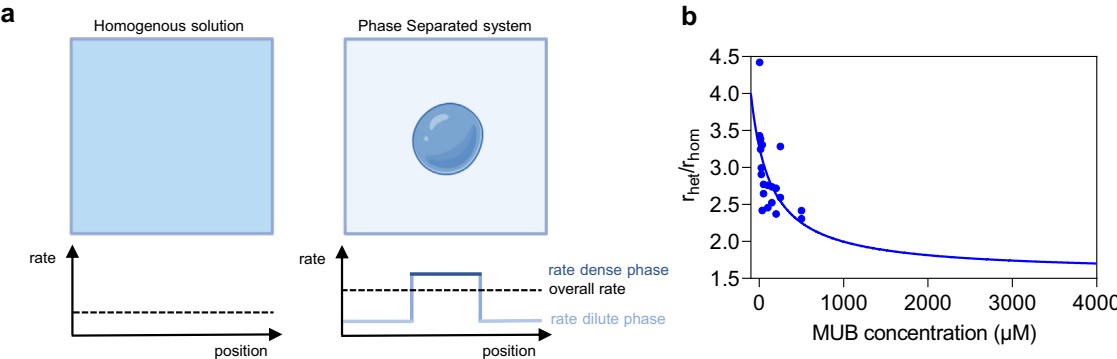

Fig. 2 | Kinetic model analysis. a Schematic illustration of local rate enhancement within condensates and overall rate of the heterogenous system, compared to the rate in a fully homogenous system. b Ratio of the rate measured in the heterogenous system ($r_{het}$) and homogenous system ($r_{hom}$) as a function of the MUB substrate concentration. The solid line represents the fitting according to Eq. 1. Source data is provided as a Source Data file. Panel (a) created in BioRender. Stoffel, F. (2025) https://BioRender.com/ha4nxuo.

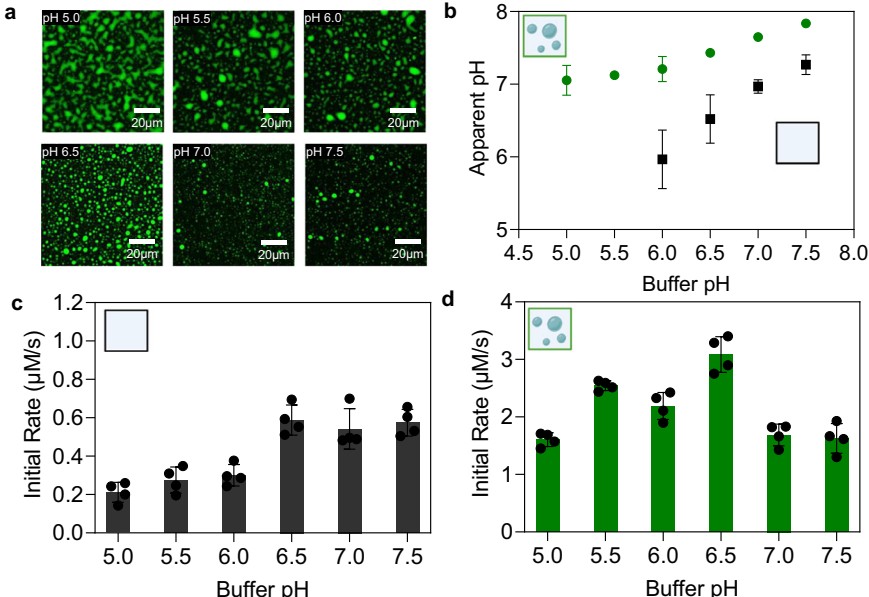

Fig. 3 | Biomolecular condensates alter the dependence of enzymatic rate on changes in solution pH. a Representative fluorescence confocal microscopy images of DDX4-BTL2-DDX4 condensates stained with SNARF−1 dye at different pH values (0.5 μM enzyme, in Tris or BisTris buffers with 10 mM ionic strength, 30 mM NaCl, 25 μM SNARF1). Fluorescence intensity was maximized for visualization purposes. b Apparent pH as measured by the SNARF-1 assay in the dense (green) and dilute (black) phases of DDX4-BTL2-DDX4 samples. (pH 5.0: 6 replicates, pH 5.5: 9 replicates, pH 6: 16 replicates, pH 6.5–7.5: 18 replicates). Error bars represent the standard error of the mean. Conditions are the same as in panel a. c, d Initial rates of BTL2 (c) and DDX4-BTL2-DDX4 (d) measured at 0.5 μM enzyme and 0.25 mM MUB in Tris/ Bis-Tris buffer at 10 mM ionic strength, 30 mM NaCl, and different pH values (4 replicates each). Error bars denote the standard error of the mean. Source data for panels b–d are provided as a Source Data file. Panel b and c created in BioRender. Stoffel, F. (2025) https://BioRender.com/sfpppex; Panel b and d created in BioRender. Stoffel, F. (2025): https://BioRender.com/ih21vl9.

as DDX4-BTL2-DDX4, is shown in Supplementary Fig. 1. The pI of DDX4-BTL2-DDX4 (6.15) is closer to the pI of BTL2 (6.70) compared to that of Laf1-BTL2-Laf1 (8.77), enabling a better evaluation of the effect of the local change of pH and of the comparison between heterogeneous and homogeneous systems. Condensates based on this construct form across a broad range of pH values (Fig. 3a) and induce a similar increase in reaction rate as Laf1-BTL2-Laf1 (Supplementary Fig. 11).

We characterized the apparent pH within DDX4-BTL2-DDX4 condensates with the same SNARF-1 assay previously applied for the Laf1-BTL2-Laf1 system. This analysis showed a significant shift of the apparent pH within the condensates towards more basic values compared to the dilute phase buffer across all tested pH values (Fig. 3b). The pH in the dilute phase could not be characterized below pH 6.0, as this exceeds the dynamic range of SNARF-1 (Supplementary Fig. 10).

Previous investigations have elucidated that the activity of BTL2 is optimal between pH 7 and 9, and drastically drops at lower pH values[36,54,55]. We therefore predicted that the more basic environment inside the condensates would increase the overall reaction rate even in more acidic buffer conditions. Indeed, we observed that the homogeneous system exhibited the expected drop in activity at lower pH values (Fig. 3c), while the enzymatic activity of the heterogeneous system was maintained or even increased in the entire pH range from 5.0 to 7.5 (Fig. 3d). The initial increase in activity with decreasing pH can be explained by the increase in the total volume fraction of the dense phase with decreasing pH (Fig. 3a and Supplementary Fig. 12). The promotion of condensation at lower pH, along with the rate-

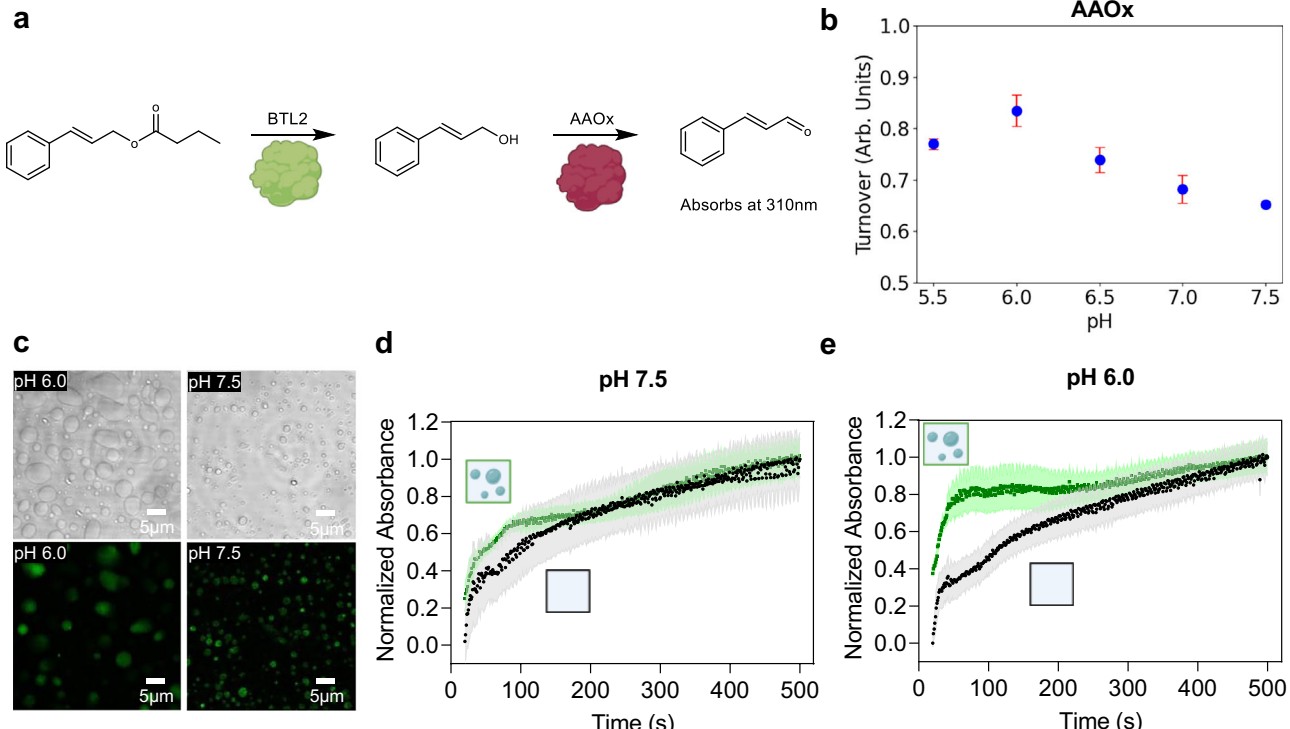

**Fig. 4 | Biomolecular condensates optimize cascade reactions. a** Cascade reaction from Cinnamyl Butyrate to Cinnamyl Aldehyde, with Cinnamyl Alcohol as an intermediate, catalyzed by BTL2 and AAOx. **b** pH dependency of AAOx activity (5 nM AAOx, 250 µM CAlc) measured after 2 mins from three independently prepared samples. Error bars denote the standard error of the mean. **c** Brightfield and fluorescence confocal images of 0.5 µM DDX4-BTL2-DDX4 condensates supplemented with 5 nM AAOx-ATTO565. **d, e** Kinetic curves of the cascade reaction performed at pH 7.5 (**d**) and at pH 6.0 (**e**) in the homogenous BTL2 (black) and heterogenous DDX4-BTL2-DDX4 (green) systems, recorded by monitoring the absorbance of the final product at 310 nm. Experiments were performed with 5 independently prepared samples each at 5 nM AAOx, 0.5 µM BTL2 or DDX4-BTL2-DDX4, 0.25 mM CB. All experiments were performed in 10 mM ionic strength Tris or Bis Tris with 30 mM NaCl. Error bars denote the standard error of the mean. Source data for panels **b**, **d** and **e** are provided as a Source Data file. Panels **d** and **e** created in BioRender. Stoffel, F. (2025): https://BioRender.com/sfpppex; panels **d** and **e** created in BioRender. Stoffel, F. (2025): https://BioRender.com/ih21vl9; panel **a** created in BioRender. Stoffel, F. (2025): https://BioRender.com/an1iv85.

enhancing effects of the condensate environment discussed earlier, likely works in conjunction with pH buffering within the condensates in maintaining a high enzymatic activity in the pH range from 5.0 to 7.5.

**Biomolecular condensates optimize cascade reactions by local pH buffering**

The results discussed in the previous paragraphs have shown that condensation can modify the response of reaction rates to changes in solution pH. This important possible function of biomolecular condensates has several implications. First of all, as shown in Fig. 3d, it can broaden the pH interval in which enzymes exhibit high activity, improving the performance of the enzyme across different environments. Moreover, this extension of the optimal pH interval opens the possibility to perform network reactions with other enzymes operating at pH values that would normally be incompatible with the reference enzyme.

We demonstrated this concept by developing a one-pot cascade reaction with a second enzyme characterized by a different optimal pH compared to BTL2. Specifically, we considered the two-step reaction from Cinnamyl Butyrate (CB) to Cinnamyl Aldehyde (CAldh) catalyzed by BTL2 and Aryl Alcohol Oxidase (AAOx) from *Pleurotus eryngii* (Fig. 4a). CB is first hydrolyzed by BTL2 to Cinnamyl Alcohol (CAlc), which is then oxidized by AAOx into CAld (Fig. 4a). The full mechanism of the oxidation reaction by AAOx is shown in Supplementary Fig. 13.

Since the histidine residue at the active site must act as a base in the reductive step of the reaction, the activity of AAOx is optimized at a slightly acidic pH value of approximately 6.0[56–58]. This was confirmed by measuring the rate of the oxidation reaction of CAlc to CAld catalyzed by AAOx between pH 5.5 and pH 7.5. Representative kinetic profiles are shown in Supplementary Fig. 13. The optimal pH for AAOx was indeed 6.0 (Fig. 4b). In contrast, the activity of BTL2 decreases at pH 6.0 compared to higher pH values (Fig. 3c).

We then added AAOX to a heterogenous DDX4-BTL2-DDX4 system, observing no significant effect on phase separation (Fig. 4c). By confocal fluorescence microscopy, we measured a partition coefficient of AAOx inside the dense phase of 156-fold at pH 6.0 and 168 at pH 7.5 (Fig. 4c, Supplementary Fig. 14). However, since the total volume fraction of the dense phase is very low (Supplementary Fig. 12), the vast majority of AAOx (approximately 98%) is still in the dilute phase, where it experiences the optimal pH of 6.0. In contrast, the lipase is mainly sequestered within the condensates, where it can operate at more basic pH values. Therefore, we expected a better performance of the cascade reaction at pH 6.0 in the DDX4-BTL2-DDX4 phase separated system compared to the homogeneous solution.

We performed the full cascade reaction at pH 6.0 and, as a reference, at pH 7.5. As shown in Fig. 4d, e the reaction in the homogeneous system containing BTL2 and AAOx is slower at pH 6.0 than pH 7.5. Since the activity of BTL2 is lower at more acidic pH values (Fig. 3c), this indicates that the hydrolysis reaction becomes rate limiting at pH 6.0. In contrast, in the phase-separated DDX4-BTL2-DDX4 system we observed a higher reaction rate at pH 6.0 than pH 7.5 (Fig. 4d, e) due to the higher activity of BTL2 in the phase separated system (Fig. 3d).

Therefore, these results demonstrate that biomolecular condensates can optimize cascade reactions by generating two aqueous

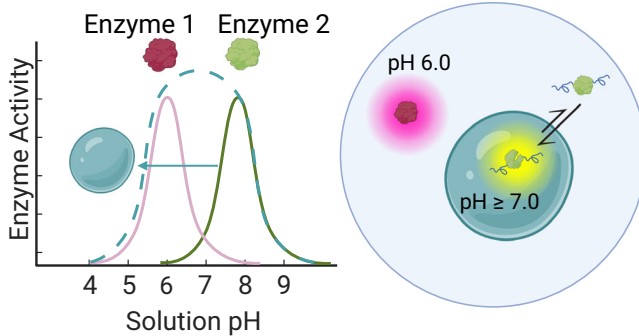

**Fig. 5 | Condensates can optimize enzymatic cascade reactions by locally changing pH.** Enzymes exhibiting activity at different pH values can be spatially segregated in the dilute and dense phase of the condensates, where they experience different pH values. Condensates act as a different effective solvent compared to the surrounding dilute phase, optimizing enzymatic activity by locally changing pH and, in the case of BTL2, by inducing conformational changes within the dense phase. Created in BioRender. Stoffel, F. (2025): https://BioRender.com/4fjnd9r.

phases characterized by distinct pH values that are optimal for different enzymes. Specifically, in this system, by sequestering the lipase in the more basic dense phase, the second step of the reaction can occur in the dilute phase at the optimal pH=6.0 for AAOx, without losing BTL2 activity.

## Discussion

Here we have designed enzymatic condensates consisting of a lipase which can undergo a conformational re-arrangement from a closed, inactive state to an open, active state upon changes in polarity of the environment and the concentration of the enzyme[39,40,42]. Since condensates exhibit a lower polarity than water and a high local enzyme concentration (in the mM range), we predicted and confirmed that the activity of the lipase increases inside the dense phase (Fig. 1).

Changes in enzyme conformations within the dense phase of condensates have already been observed with the enzyme superoxide dismutase, which was shown to be preferentially unfolded in the dense phase of condensates[34]. Moreover, the mRNA decapping complex, Dcp1/Dcp2, which exists in a conformational equilibrium between active and inactive states, shifts towards the inactive state within condensates, subsequently impacting RNA decapping[35]. These studies, together with the results shown in this work, suggest that modulation of enzyme conformation can represent a widespread mechanism underlying regulation of enzymatic activity by condensation.

In addition to changing enzymatic activity, we have shown that condensates can locally modify the pH compared to the surrounding solution, in agreement with recent studies[27,48–51]. By locally inducing a more basic environment, where lipase activity is favored, biomolecular condensates could thereby increase the lipase activity over a much broader pH range compared to the homogeneous solution (Fig. 3c, d).

These results have several important implications, including the possibility to optimize cascade reactions of different enzymes that require distinct optimal pH values. Here we have demonstrated this concept by optimizing a two-step cascade reaction catalyzed by BTL2 and AAOx (Fig. 4). Since most of the BTL2 is sequestered inside the more basic condensates, while the vast majority of AAOx is in the dilute phase, the reaction can be performed at acidic pH values that are optimal for AAOx, without compromising the activity of BTL2 (Fig. 5). This mechanism is reminiscent of the nanobuffering effect observed with the conjugation of enzyme with synthetic polymers that can locally modify the pH around the protein[59,60], which was employed to optimize a cascade reaction in bulk[59].

Previous coacervate-based approaches designed to optimize cascade reactions have largely focused on co-localization of enzymes

in two or three compartments, to minimize mass transport limitations or substrate inhibition, induce proximity, and provide directionality in space[17,61,62]. Our approach shows an additional benefit of open compartmentalization, namely the use of the distinct physiochemical properties of the different phases to optimize otherwise incompatible reactions.

Cascade reactions based on enzymes that require different pH values typically require immobilization or use of whole cells[63]. Biomolecular condensates represent a simpler approach, enabling the assembly of enzymatic cascades in one pot using purified enzymes. This is of particular interest in the case of lipases, whose hydrophobic substrates can exhibit significant mass transfer limitations through the cellular membrane when used in whole-cell biocatalysts, or adhere to microparticles used for enzyme immobilization[64].

Our results, together with previous findings[15,18,30], show that condensates can modulate biochemical activity by acting as effective solvents that are distinct from the surrounding phase. In addition to changes in the local concentrations, important effects of biomolecular condensates on enzymatic activity include local variations in effective polarity[13,43], water thermodynamics[32], enzyme conformation[34,35], mass transfer limitations, and the presence of an interface[26–28]. These effects are convoluted[65], and can result in a dependence of the enzymatic rate on the size of the condensates, as recently shown by experiments[15] and computations[66]. A particularly interesting effect is the local change of pH[27,48–51], due to the partitioning of charged ions between the dilute and dense phase. This generates an interphase electric potential, which can lead to catalytic activity[26–28].

Here, we demonstrate an additional consequence of the different pH value within biomolecular condensates, namely the possibility to broaden the interval of solution pH values in which enzymes exhibit high activity and increase the robustness of enzymatic rates upon changes in external parameters. This pH buffering represents a possible function of biological enzymatic condensates and opens attractive applications of synthetic condensates in biocatalytic engineering and optimization of cascade reactions.

## Methods
### Protein expression and purification

cDNAs encoding for the intrinsically disordered domains of Laf1 (indicated in the following as Laf1 IDR), Laf1-BTL2-Laf1, BTL2 and AAOx proteins were cloned into a pET-15b vector by Genewiz, Azenta Life Sciences (Leibzig, DE) including an Ampicillin resistant region and an N-terminal His-tag in the POI region. DDX4-BTL2-DDX4 was cloned from Laf1-BTL2-Laf1 using Golden Gate cloning. The full sequences of the constructs are shown in Supplementary Fig. 1. Proteins were expressed recombinantly in BL21-GOLD *E.coli* cells in enriched LB Media (Laf1-BTL2-Laf1, DDX4-BTL2-DDX4) or LB Media (Faust Laborbedarf AG, 6271000) (Laf1, BTL2, AAOx) in the presence of 100 μg/mL Ampicillin (Applichem, A0839). Protein expression was induced at an OD of 0.5 (Laf1-BTL2-Laf1, BTL2, DDX4-BTL2-DDX4, Laf1 IDR) or 1.0 (AAOx) with 0.5 mM IPTG (99%, Applichem, A1008). For the expression of AAOx, cells were cold shocked for 30 min prior to induction. Cells were grown for an additional 16 h at 37 °C (for Laf1 IDR), 20 °C (for BTL2) or 16 °C (for Laf1-BTL2-Laf1, DDX4-BTL2-DDX4, and AAOx) before harvesting. Soluble proteins were extracted via cell sonication and centrifugation at 12,800 rpm for 30 min. For Laf1 IDR, the lysis buffer was supplemented with 8 M urea to extract the protein from inclusion bodies. Target proteins were initially purified by immobilized metal ion affinity chromatography (Chelating Sepharose, GE Healthcare) following a standard protocol. During the purification of AAOx, 0.08 mM FAD was added prior to cell lysis and after IMAC elution. For all proteins, the IMAC eluent was further purified by Size Exclusion Chromatography using a Superdex Hiload16/600 200 pg column (Cytiva) assembled on an Akta prime system (GE Healthcare). For the Laf1 IDR protein, a 75 pg resin was used. The eluent SEC buffers were

optimized for the different proteins: for BTL2, 50 mM Tris, 500 mM NaCl, pH 8.5; for Laf1-BTL2-Laf1, 50 mM Tris, 750 mM NaCl, 10% glycerol, pH 9.0; for DDX4-BTL2-DDX4, 50 mM Tris pH 8.5, 1 M NaCl, 10% glycerol; for AAOx, 50 mM Tris, pH 7.5, 500 mM NaCl, 10% glycerol; for Laf1 IDR, 50 mM Tris, pH 7.5, 800 mM NaCl, 2 M urea). Protein purity was verified by SDS-PAGE. The target protein was concentrated to approximately 450 μM for BTL2 and Laf1 IDR, 30 μM for Laf1-BTL2-Laf1 and AAOX and 17 μM for DDX4-BTL2-DDX4. Proteins were stored at −80 °C as aliquots. Protein concentrations were measured by UV absorbance at 280 nm using a Nanodrop Lite Spectrophotometer by Thermo Scientific.

### Polarity measurement using PRODAN

Phase-separated Laf1-BTL2-Laf1 samples were prepared in 24 mM Tris Buffer, 10 mM NaCl, pH 7.5 at a concentration of 0.5 μM in a MicroWell Glass Bottom MatriPlate by Azenta Life Sciences. Prodan (Sigma, 41,525) was added at final concentration of 100 μM. Fluorescence signal was recorded between 410 nm and 600 nm after excitation at 405 nM in 5 nm increments using a confocal microscope (Leica TCS SP8-AOBS, Hamamatsu Orca Flash 4.0, sCMOS camera; emission detection with a PMT2 detector). The fluorescence spectrum was extracted by image analysis using homemade python scripts. The wavelengths corresponding to the maximum signal for isopropanol and water were measured using a fluorescence detection platereader (CLARIOstar, BMG Labtech, Ortenberg, Germany) with the same excitation/emission settings.

### Analysis of enzyme partitioning via size exclusion chromatography

To determine the protein concentration in the dilute phase, 0.5 μM Laf1-BTL2-Laf1 protein solution was prepared in 24 mM Tris buffer at pH 7.5 and 10 mM NaCl. The dilute phase was separated from the dense phase by centrifugation at 4500 rpm for 30 min. 40 μL of the supernatant was then injected into a Superdex 200 Increase 5/150 GL (Cytiva) size exclusion column, connected to an Agilent 1200 Series HPLC. As a running buffer, 50 mM Tris with 750 mM NaCl at pH 7.5 was used. The protein concentration was recorded using am 1260 Infinity II FLD spectrophotometer. The measurement was repeated at least four times with separately prepared samples. FLD data was analyzed using a python code: the spectra were baselined and the area under the peak of interest calculated. As reference, a homogeneous solution of 0.5 μM Laf1-BTL2-Laf1 was prepared in 50 mM Tris at pH 9.0 with 750 mM NaCl and 10% glycerol, and analyzed at least four times with independent samples following the same protocol. The fraction of protein remaining in the dilute phase was determined as the fraction of the area of the dilute phase samples divided by the area of the homogeneous sample.

### Measurement of total volume fraction of the dense phase

Condensates were prepared at a protein concentration of 0.5 μM in 24 mM Tris, pH 7.5, 10 mM NaCl for Laf1-BTL2-Laf1, and 30 mM NaCl for DDX4-BTL2-DDX4. Samples were prepared in a MicroWell Glass Bottom MatriPlate (Azenta Life Sciences) and centrifuged for 30 min at 4000 rpm. Condensates were stained with 25 μM SNARF-1 (Thermo-Fisher Scientific, C1270). The z-stack images were recorded using a confocal microscope (Leica TCS SP8-AOBS, Hamamatsu Orca Flash 4.0, sCMOS camera; excitation: 488 nm, Argon Laser 20%, emission detection with a PMT2 detector: 625-680 nm). Images were recorded at 0.1 μm intervals and analyzed by home-made codes to extract the volume fraction. At least three individually prepared samples were imaged.

### Kinetics of MUB hydrolysis by lipase

Solutions of 4-Methyl Umbelliferyl Butyrate (Sigma, 19362) and enzymes in the desired buffer were mixed at a 1:1 ratio in a 96-well, half area, polystyrene flat bottom plate (Corning). The sample volume was 40 μL. The fluorescence signal of the product was recorded over time using a fluorescence detection platereader (CLARIOstar, BMG Labtech, Ortenberg, Germany), using an excitation wavelength of 360 nm and measuring emission at 450 nm. Measurements were taken every 5 s, with the first measurement taken 25 s after mixing, shaking for 1 s at 500 rpm between measurements. For the Michaelis Menten analysis at least four separately prepared samples were analyzed per substrate concentration. A Methyl Umbelliferone (Sigma, M1381) calibration curve was evaluated for each buffer condition (see Supplementary Fig. 15). The effect of the Laf1 intrinsically disordered domain was analyzed on a Spark plate reader (Tecan Trading AG, Switzerland) with sample preparation and measurements as described above.

### Resorufin partitioning into Laf1-BTL2-Laf1 condensates

Condensates were prepared in a MicroWell Glass Bottom MatriPlate (Azenta Life Sciences) at a protein concentration of 0.5 μM in 24 mM Tris Buffer, pH 7.5, 10 mM NaCl, and 50 μM Resorufin (Fluorochem, M03613). Condensates were allowed to sediment before imaging using a confocal microscope (Leica TCS SP8-AOBS, Hamamatsu Orca Flash 4.0, sCMOS). Fluorescence emission was recorded in the range 564 nm–616 nm after excitation at 561 nm (PMT detector, Gain 600 V). The intensity profile of 30 condensates from 6 different wells was analyzed using ImageJ.

### Fluorescence correlation spectroscopy

Laf1-BTL2-Laf1 condensates were prepared in a MicroWell Glass Bottom MatriPlate (Azenta Life Sciences) at a concentration of 0.5 μM. Atto-565 (Sigma, 75784-1MG-F) was added with a final concentration of 20 nM. FCS experiments were performed using an inverted confocal fluorescence microscope (Leica SP8 STED) equipped with an HC PL APO CS2 63 × 1.2 NA water immersion objective with software-controlled correction collar (Leica) and a hybrid detector for single molecule detection (HyD SMD). Data acquisition and analysis were carried out using Leica Application Suite X software (version 1.0). The confocal volume was calibrated using Atto-565 NHS Ester (Diffusion coefficient = 400 μm²/s), yielding an effective volume ($V_{eff}$) of 0.312 fl and a focal volume height-width ratio (K) of 5.871. Samples were excited with a 555 nm laser from a White Light Laser at 80 MHz repetition frequency, and the fluorescence emission was collected within the 570–600 nm wavelength range. The pinhole size was 100 μm. The diffusivity of the Atto 565 dye within Laf1-BTL2-Laf1 condensates was determined from the autocorrelation curve.

### pH measurement using SNARF-1

Condensates were prepared in a MicroWell Glass Bottom MatriPlate (Azenta Life Sciences) at a concentration of 0.5 μM in Tris / Bis Tris buffers with ionic strength 10 mM (Laf1-BTL2-Laf1, DDX4-BTL2-DDX4). SNARF-1 (Thermo Fisher Scientific, C1270) was added at a concentration of 25 μM. Condensates were allowed to sediment before analyzing spectral properties by excitation at 488 nm and measurement of the two emission maxima (570 nm–590 nm and 630 nm–650 nm) using a confocal microscope (Leica TCS SP8-AOBS, Hamamatsu Orca Flash 4.0, sCMOS, PMT detectors, Gain 600 V) and an argon laser power source. The calibration curve was constructed for buffers from pH 6.0–9.0 at 0.5 pH intervals. For the calibration curve, the final SNARF-1 concentration was 50 μM. The intensity signals at the two maximum wavelengths was extracted from image analysis using home-made codes.

### AAOx labeling with ATTO565

Fresh AAOx protein was purified as described above, with the only difference being the elution buffer of the SEC step: 1 x PBS, at pH 7.5 with 350 mM NaCl and 10% glycerol. ATTO-565 NHS Ester (Supelco, 72,464) was added at a 2:1 dye: protein ratio, and the sample was incubated at room temperature for two hours. Free dye was then

removed by SEC using a Superdex Increase 10/300 75 pg column (Cytiva) assembled on an Akta Pure system (Cytiva), using 50 mM Tris, 500 mM NaCl, 10% Glycerol, pH 7.5 as elution buffer. The purified labeled protein was concentrated to 2 μM and stored at −80 °C as aliquots. Protein concentration was measured via UV absorbance at 280 nm and 565 nm using a Nanodrop One Spectrophotometer (Thermo Scientific).

### AAOx partitioning into DDX4-BTL2-DDX4 condensates

Condensates were prepared in a MicroWell Glass Bottom MatriPlate (Azenta Life Sciences) at a concentration of 0.5 μM. 5 nM AAOx-ATTO565 was added to the sample and uptake was measured by excitation at 561 nm using a confocal microscope (Leica TCS SP8-AOBS, Hamamatsu Orca Flash 4.0, sCMOS). Emission was monitored with a PMT detector (585 nm–620 nm). For each buffer condition, three distinct wells were imaged and at three condensates were analyzed using ImageJ (Intensity Profile).

### AAOx activity assay

Solutions of Cinnamyl Alcohol (Sigma, 108197-5 G) and AAOx in the desired buffer were mixed at a 1:1 ratio in a MicroWell Glass Bottom MatriPlate (Azenta Life Sciences). The absorbance of the product (Cinnamyl Aldehyde, Sigma, W228613) was monitored over time at 310 nm using a platereader (CLARIOstar, BMG Labtech, Ortenberg, Germany). Measurements were performed at a 1 s interval, with the first measurement taken 20 s after mixing. All measurements were performed in independent triplicates. The same procedure was applied to monitor the cascade reaction with the BTL2 enzyme, where the enzyme mixture was prepared prior addition of the Cinnamyl Butyrate (TCI, C2438) substrate as described above. All measurements were performed with five independent replicates. For data normalization, the background signal of BTL2 or DDX4-BTL2-DDX4 protein in the same buffer was used.

### Reporting summary

Further information on research design is available in the Nature Portfolio Reporting Summary linked to this article.

## Data availability

Unless otherwise stated, all data supporting the results of this study can be found in the article, supplementary, and source data files. The processed data for all plots in this study are available in the Source Data file. Raw microscopy data for volume fraction analyses, PRODAN or Resorufin analyses are available from the corresponding author upon request due to large file sizes. The PDB used for the open conformation of BTL2 is 2w22, the PDB used for the closed conformation is 1ji3 (analog from *G. Stearothermophilus*) [https://doi.org/10.2210/pdb1JI3/pdb]. Source data are provided with this paper.

## Code availability

Custom codes used for analysis are available https://doi.org/10.5281/zenodo.15340756.

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

## Acknowledgments

We kindly acknowledge the European Research Council through the Horizon 2020 research and innovation program (grant agreement No. 101002094) (PA) for financial support. The authors thank Prof. Benjamin Schuler (University of Zurich) for scientific discussions.

## Author contributions

P.A. and F.S. designed the conceptual framework of the study. F.S. performed the experiments. M.P. performed the kinetic model analysis. A.M.K., A.I.B.-M., R.J., M.G.-G., N.G., L.F., and P.A. contributed to data acquisition and interpretation. P.A. supervised the project. F.S. and P.A. wrote the manuscript with contributions from all authors.

## Funding

## Competing interests

The authors declare no competing interests.
