## [Transparent Peer Review file · Nature Communications]

Enhancement of Enzymatic Activity by Biomolecular Condensates through pH Buffering

Corresponding Author: Professor Paolo Arosio

Version 0:

Reviewer comments:

Reviewer #1

(Remarks to the Author)

Biomolecular condensates enhance enzymatic reactions by altering local molecule concentrations and creating distinct microenvironments. Using a lipase from *Bacillus thermocatenulatus*, the authors show that phase separation boosts enzyme activity by stabilizing its active conformation and creating a basic microenvironment, maintaining activity even in unfavorable bulk pH conditions. Additionally, condensates optimize cascade reactions by buffering pH locally, enabling enzymes with different pH optima to function together. Their findings highlight the potential of condensates in biocatalytic engineering to expand reaction conditions and improve robustness.

The paper is well written, the findings are new and supported by the data. However the below corrections (major and minor) must be addressed before publication:

1. in the intro a few refs on the enzymatic reactions in MO are missing. There are plenty in literature that can be cited here.
2. "Biomolecular condensates form via phase separation of proteins and nucleic acids", this is most of the cases but we have condensate constituted only by proteins.
3. "Like all members of the lipase family³⁸," I do not think all the lipase has the lid subdomain, I think most of the lipase has the lid.
4. somewhere in the intro this paper must be cited: I think this paper must be cited: <https://www.nature.com/articles/s41586-023-06626-z>
5. "To further support this finding, we measured the enzymatic rate at 10 mM NaCl and low enzyme concentration (20 nM), where no condensates form (Suppl Fig. S3). Again, no difference in reaction rate was observed between the BTL2 and Laf1-BTL2-Laf1 systems (Figure 1H)." To further confirm this result, it would be interesting to repeat the experiment of panel 1F using Laf1-BTL2-Laf1 where no increase in activity in the presence of 10% isopropanol is expected.
6. in figure S5 the trend is clear. However, in the panel A of figure S5, the author compare the activity of Laf1 and BTL2 in different buffer conditions (Laf1 contains different NaCl concentration and the presence of 30 mM urea). Since they are placed in the same graph, I would test them in the same buffer conditions.
7. It is needed to have a supporting figure showing the kinetic curves (fluorescence vs time) of all the different substrate concentrations from which these data are derived.
8. Figure 3: To clarify whether the pH effect observed in panel B is due to the hydrophobic environment of the condensate or the pH itself—since both factors are hypothesized to influence BTL2 catalysis—it is recommended to use Laf1-BTL2-Laf1 as a negative control. Including this construct in all experiments for Figure 3 will help separate the effects of the condensate's hydrophobicity from its pH, providing a clearer understanding of their respective contributions to enzymatic activity.
9. "The pH in the dilute phase could not be characterized below pH 6.0, as this exceeds the dynamic range of the SNARF-1 probe (Suppl. Fig. S7).

Previous investigations have elucidated that the activity of BTL2 is optimal between pH 7 and 9, and drastically drops at lower pH values.^{37,54,55}

I understand that pH characterization with SNARF is limited below pH 6 but can detect changes up to pH 9. Since the enzyme's optimal pH range is between 7.0 and 9.0 (as mentioned here), this range has to be tested to provide a more complete understanding of its activity within its optimal pH conditions.

Reviewer #2

(Remarks to the Author)

In this work, Arosio and coworkers leverage their previous designs of enzyme-IDR fusions to show that biomolecular condensates enhance the enzymatic rate of the lipase BTL2 by locally modifying the pH within the dense phase. These findings might be further used to expand the optimal pH range and improve the robustness of enzymatic reactions under varying environmental conditions. Additionally, the authors showed that the pH buffering which results in a different pH in each of the phases in the heterogeneous solution, enables a cascade reaction of enzymes which do not work optimally in the same pH.

I find this work very interesting. The strongest part of the manuscript is the derivation of the Michaelis-Menten kinetic model, which is very helpful, and the analysis of the local pH in the dense phase, as well as the enzymatic cascade demonstration. Yet, I think that the paper needs some additional work (not all of it should require additional experiments), to be suitable for publication in *Nat Commun*. A few critical points are the LLPS analysis part, which is a bit neglected here, the selection of the substrate, which requires some clarification, and there are a few claims which are not supported experimentally and require revision – please see my comments below. After a major revision I believe that the manuscript can be suitable for publication as it is very relevant to the LLPS community.

- 1) Figure 1A: it is a bit difficult to observe the closed and open conformation of the enzyme by this overlapped schematic. I recommend separating the two 3D structures.
- 2) Figure 1B: From the schematic illustration it seems that the IDR-enzyme fusion construct is a payload and does not form the droplet. Please revise the schematics to more clearly show that the construct undergoes LLPS into condensates.
- 3) The LLPS propensity of the construct is higher at acidic pH. Please elaborate on this, as it is not typical for other protein or peptide-based LLPS systems.
- 4) “We therefore next assessed whether the environment of biomolecular condensates can increase enzymatic reaction similarly to organic solvents”: this line is unclear as typically organic solvents restrict the catalytic activity of enzymes – please explain.
- 5) Which reaction exactly is catalyzed by the selected enzyme? there is no mentioning of which group or bond is being cleaved (and no mentioning of hydrolysis in the beginning of the results section). Figure 1F shows ester hydrolysis reaction. Is this the common reaction catalyzed by BTL2?
- 6) A follow-up comment: it is clear why the authors have selected MU-based system due to the ability to track the reaction, but I am wondering about the relevancy of the selected substrate as an enzymatic model system for lipase, as the aliphatic chain of MUB is so short. Please explain this selection and provide some literature support of lipase activation on such short fatty acids.
- 7) The kinetics analysis (table 1) shows a very moderate increase in the V_{max} of the heterogeneous system ($3.8 \mu\text{M s}^{-1}$) compared to the free enzyme ($2.1 \mu\text{M s}^{-1}$). Can the authors explain why the threefold enhancement effect observed for the initial rate of the heterogeneous system vs. the homogeneous system is not observed for V_{max} ?
- 8) I am not sure that the analysis with 750 mM NaCl demonstrates that there is no enhancing effect of the LAF1 fusion, as at this high ionic strength the enzyme activity is probably hindered.
- 9) The analysis confirming that the LAF1 condensates alone are not catalytic (Fig. S5) is important and I think that the authors wisely included it.
- 10) How did the authors overcome potential changes in the emission intensity of the fluorescent product in the heterogeneous system due to scattering? it is worthwhile mentioning it in the text.
- 11) Page 7 kinetic model: the authors state that the evaluation of the activity coefficient in the dense phase requires assumption about the diffusion of the substrate. Since MUB is a fluorescent molecule, can the authors analyze its diffusion in the dense phase by FRAP?
- 12) Do the authors know experimentally what is the partitioning coefficient of the product?
- 13) The authors state that the kinetics analysis supports their initial hypothesis about conformational changes of the enzyme in the dense phase. While I agree that this is very likely to happen, there is no direct experimental evidence that reaffirms this hypothesis. I suggest that the authors should either re-phrase this statement or perform additional biophysical experiments to directly analyze the conformation of the enzyme in different viscosities/conditions that can mimic the dense phase.

Minor points:

The figures include a lot of white spaces. Although not critical, I would recommend re-organizing the panels in Figure 1 to avoid these spacings.

Version 1:

Reviewer comments:

Reviewer #1

(Remarks to the Author)

The authors have made substantial improvements to the manuscript by thoughtfully addressing the concerns raised in my previous review. The manuscript is now clear and robust.

Reviewer #2

(Remarks to the Author)

The authors have fully addressed my concerns and comments. I appreciate their effort in re-validating their kinetic parameters and confirming that the fluorescence of the product is not hindered in coacervates as well as the FCS diffusion analysis, which strengthen the manuscript. The manuscript is now suitable for publication.

Reviewer #1 (Remarks to the Author)

Biomolecular condensates enhance enzymatic reactions by altering local molecule concentrations and creating distinct microenvironments. Using a lipase from *Bacillus thermocatenuatus*, the authors show that phase separation boosts enzyme activity by stabilizing its active conformation and creating a basic microenvironment, maintaining activity even in unfavorable bulk pH conditions. Additionally, condensates optimize cascade reactions by buffering pH locally, enabling enzymes with different pH optima to function together. Their findings highlight the potential of condensates in biocatalytic engineering to expand reaction conditions and improve robustness.

The paper is well written, the findings are new and supported by the data. However the below corrections (major and minor) must be addressed before publication:

Answer from the authors: We thank the reviewer for the overall positive feedback and the constructive comments, which we have addressed as described in the point-to-point responses below.

1. in the intro a few refs on the enzymatic reactions in MO are missing. There are plenty in literature that can be cited here.

Answer from the authors: We have now added references on studies connecting biomolecular condensates with regulation of gene expression, promotion of mRNA translation, regulation of metabolic flux, and the pyrenoid (references 11-18).

Despite the increasing number of observations correlating enzymatic reactions with MOs¹¹⁻¹⁸, the molecular mechanisms underlying the potential role of biomolecular condensates in enzymatic reactions remain largely unclear.

2. "Biomolecular condensates form via phase separation of proteins and nucleic acids", this is most of the cases but we have condensate constituted only by proteins.

Answer from the authors: Prompted by the reviewer's comment we adjusted this section to include both homotypic and heterotypic condensates with accompanying references.

Biomolecular condensates form via phase separation of proteins,^{19,29,30} often in combination with nucleic acids^{31,32}, resulting in the formation of a high-concentration phase surrounded by a lower-concentration phase.

3. "Like all members of the lipase family³⁸," I do not think all the lipase has the lid subdomain, I think most of the lipase has the lid.

Answer from the authors: We thank the reviewer for noticing this. We have adjusted the statement in question to avoid overgeneralization.

Like many members of the lipase family⁵⁰, BTL2 can transition from a closed, inactive state to an open, active state by extensive rearrangement of the lid domain, which results in the exposure of a hydrophobic binding pocket (Figure 1A).⁵¹

4. somewhere in the intro this paper must be cited: I think this paper must be cited:

<https://www.nature.com/articles/s41586-023-06626-z>

Answer from the authors: We thank the reviewer for this suggestion. The reference has been added to the introduction as ref 45:

Alongside variations in local concentrations, emerging evidence suggests that biomolecular condensates can exert other significant effects. These include the formation of an interface between the dense and dilute phase^{21,31,39-42} and the change of local environment within the dense phase compared to the dilute phase^{19,21,38,43,44}, such as water thermodynamics⁴⁵.

5. "To further support this finding, we measured the enzymatic rate at 10 mM NaCl and low enzyme concentration (20 nM), where no condensates form (Suppl Fig. S3). Again, no difference in reaction rate was observed between the

BTL2 and Laf1-BTL2-Laf1 systems (Figure 1H).” To further confirm this result, it would be interesting to repeat the experiment of panel 1F using Laf1-BTL2-Laf1 where no increase in activity in the presence of 10% isopropanol is expected.

Answer from the authors: We are not entirely sure about the suggestion of the reviewer. If we understand correctly, the expectation is that in the presence of 10% isopropanol there should not be any difference with / without phase separation, as the protein would have been already activated in the dense phase by the condensate environment. However, the results of this measurement would be extremely difficult to interpret, as many effects will be convoluted: the effect of phase separation could still add to the presence of isopropanol, which in turn could affect concentrations and conformations in the dense phase in a convoluted way.

6. in figure S5 the trend is clear. However, in the panel A of figure S5, the author compare the activity of Laf1 and BTL2 in different buffer conditions (Laf1 contains different NaCl concentration and the presence of 30 mM urea). Since they are placed in the same graph, I would test them in the same buffer conditions.

Answer from the authors: Prompted by the comment of the reviewer, we repeated the experiment using the same buffer (24mM Tris, 30mM NaCl, 20mM Urea, pH 7.5) with both the BTL2 enzyme and the Laf1 IDR.

Supplementary Figure 6: A) MUB hydrolysis over time in presence of BTL2 (black symbols, 0.5 μ M protein, 0.1 mM MUB) and Laf1 IDR condensates (blue symbols, 10 μ M protein, 0.1 mM MUB) in 24 mM Tris buffer at pH 7.5 and 30 mM NaCl, 30 mM Urea. BTL2 and Laf1 condensates were each measured in independent triplicates. **B)** Brightfield microscopy image of Laf1 IDR condensates before adding the substrate for the experiment shown in panel A.

7. It is needed to have a supporting figure showing the kinetic curves (fluorescence vs time) of all the different substrate concentrations from which these data are derived.

Answer from the authors: Following the reviewer’s suggestion, the following figure was added to the supplementary information:

Supplementary Figure 7: Kinetic curves of MUB hydrolysis reaction catalysed by homogenous BTL2 (**A**) and Laf1-BTL2-Laf1 condensates (**B**) at different MUB substrate concentrations (indicated in μM concentrations) in 24mM Tris, 10mM NaCl, pH 7.5. Each condition was analysed by at least four independently prepared samples. Error bars indicate standard error of the mean.

8. Figure 3: To clarify whether the pH effect observed in panel B is due to the hydrophobic environment of the condensate or the pH itself—since both factors are hypothesized to influence BTL2 catalysis—it is recommended to use Laf1-BTL2-Laf1 as a negative control. Including this construct in all experiments for Figure 3 will help separate the effects of the condensate's hydrophobicity from its pH, providing a clearer understanding of their respective contributions to enzymatic activity.

Answer from the authors: We appreciate the suggestion of the reviewer but also in this case it is not so easy to deconvolute the contributions of the individual effects. Indeed, also the Laf1-BTL2-Laf1 condensates exhibit a pH difference, in addition to the hydrophobic environment and upconcentration (Supplementary Figure 10).

We have modified the sentence to clarify this point:

The promotion of condensation at lower pH, along with the rate-enhancing effects of the condensate environment discussed earlier, likely works in conjunction with pH buffering within the condensates to maintain high enzymatic activity in the pH range from 5.0 to 7.5.

9. “The pH in the dilute phase could not be characterized below pH 6.0, as this exceeds the dynamic range of the SNARF-1 probe (Suppl. Fig. S7).

Previous investigations have elucidated that the activity of BTL2 is optimal between pH 7 and 9, and drastically drops at lower pH values.^{37,54,55}”

I understand that pH characterization with SNARF is limited below pH 6 but can detect changes up to pH 9. Since the enzyme's optimal pH range is between 7.0 and 9.0 (as mentioned here), this range has to be tested to provide a more complete understanding of its activity within its optimal pH conditions.

Answer from the authors: We appreciate the comment of the reviewer. However, we had to limit the analysis to pH 7.5 due to extensive autolysis of the MUB substrate at more basic pHs (see data below), thereby leading to a convoluted output.

Autolysis of MUB - pH dependence

Reviewer #2 (Remarks to the Author)

In this work, Arosio and coworkers leverage their previous designs of enzyme-IDR fusions to show that biomolecular condensates enhance the enzymatic rate of the lipase BTL2 by locally modifying the pH within the dense phase. These findings might be further used to expand the optimal pH range and improves the robustness of enzymatic reactions under varying environmental conditions. Additionally, the authors showed that the pH buffering which results in a different pH in each of the phases in the heterogenous solution, enables a cascade reaction of enzymes which do not work optimally in the same pH.

I find this work very interesting. The strongest part of the manuscript is the derivation of the Michaelis Menten kinetic model, which is very helpful, and the analysis of the local pH in the dense phase, as well as the enzymatic cascade demonstration. Yet, I think that the paper needs some additional work (not all of it should requires additional experiments), to be suitable for publication in Nat Commun. A few critical points are the LLPS analysis part, which is a bit neglected here, the selection of the substrate, which requires some clarification, and there are a few claims which are not supported experimentally and requires revision – please see my comments below. After a major revision I believe that the manuscript can be suitable for publication as it is very relevant to the LLPS community.

Answer from the authors: We thank the reviewer for the positive feedback and the constructive comments, which we have addressed as described in the point-to-point responses below.

1) Figure 1A: it is a bit difficult to observe the closed and open conformation of the enzyme by this overlapped schematics. I recommend separating the two 3D structures.

Answer from the authors: We thank the reviewer for this suggestion. Figure 1A has been adapted accordingly:

Created in BioRender. Stoffel, F. (2025) <https://BioRender.com/bb8rbcq>

2) Figure 1B: From the schematic illustration it seems that the IDR-enzyme fusion construct is a payload and does not form the droplet. Please revise the schematics to more clearly show that the construct undergoes LLPS into condensates.

Answer from the authors: We thank the reviewer for this suggestion. Figure 1B has been adapted to better visualize that the fusion protein is the species undergoing LLPS:

Created in BioRender. Stoffel, F. (2025) <https://BioRender.com/r5e105g>

3) The LLPS propensity of the construct is higher at acidic pH. Please elaborate on this, as it is not typical for other protein or peptide-based LLPS systems.

Answer from the authors: DDX4 is a member of the DEAD-box helicase family of proteins and enhanced phase separation at lower pHs for this family of proteins has been observed for Dhh1 by Hondele *et al.* and also in our lab for various fusion proteins using IDR from these proteins. (see references below). The mechanistic understanding of this behaviour is indeed an interesting question, however it is outside the scope of this work.

Moreover, as we are working with a synthetic fusion protein, in addition to the IDR-IDR interactions there could also be cooperative interactions between the IDR and the globular domain, which further complicates the analysis.

Hondele, M., Sachdev, R., Heinrich, S. *et al.* DEAD-box ATPases are global regulators of phase-separated organelles. *Nature* 573, 144–148 (2019). <https://doi.org/10.1038/s41586-019-1502-y>

Lenka Faltova, Andreas M. Küffner, Maria Hondele, Karsten Weis, and Paolo Arosio *ACS Nano* 2018 12 (10), 9991-9999 DOI: 10.1021/acsnano.8b04304

4) “We therefore next assessed whether the environment of biomolecular condensates can increase enzymatic reaction similarly to organic solvents”: this line is unclear as typically organic solvents restrict the catalytic activity of enzymes – please explain.

Answer from the authors: Although indeed many enzymes organic solvents limit the enzymatic activity, for many lipases the active conformation is promoted in a hydrophobic environment such as an organic co-solvent.

We have now clarified as follows:.

First, the multiple salt bridges that stabilize the open conformation can be favoured in more hydrophobic environments than water (e.g. an alcohol-water mixture).^{52–54}

5) Which reaction exactly is catalyzed by the selected enzyme? there is no mentioning of which group or bond is being cleaved (and no mentioning of hydrolysis in the beginning of the results section). Figure 1F shows ester hydrolysis reaction. Is this the common reaction catalyzed by BTL2?

Answer from the authors: The reaction catalyzed by BTL2 in this work is the ester hydrolysis depicted in Figure 1F. It is also described as such in the Results section on page 3 when the reaction is first discussed. The hydrolysis reaction is the natural function of BTL2, however esterification or transesterification reactions are also possible. These however require alternative conditions where water content is minimal and the enzyme is for example, immobilized for stabilization in organic solvent. This is described in some more detail in the following publication:

Adriano A. Mendes, Pedro C. Oliveira, Ana M. Vélez, Roberto C. Giordano, Raquel de L.C. Giordano, Heizer F. de Castro, Evaluation of immobilized lipases on poly-hydroxybutyrate beads to catalyze biodiesel synthesis, *International Journal of Biological Macromolecules*, Volume 50, Issue 3, 2012, Pages 503-511, <https://doi.org/10.1016/j.ijbiomac.2012.01.020>.

6) A follow-up comment: it is clear why the authors have selected MU-based system due to the ability to track the reaction, but I am wondering about the relevancy of the selected substrate as an enzymatic model system for lipase, as the aliphatic chain of MUB is so short. Please explain this selection and provide some literature support of lipase activation on such short fatty acids.

Answer from the authors: Natural substrates of BTL2 are mainly long-chain fatty acids which are not soluble in water, thus forming a two-phase system which would have compromised the analysis. To test our hypothesis, we selected MUB because of its solubility, which allows for the investigation of parameters such as the local environment of the enzyme.

MUB was previously used to investigate the effect of organic solvents in an aqueous BTL2 system (Shehata *et al.*) and Prim *et al.* highlighted the use of MUB as a readout methodology for general lipase characterization.

Shehata, M., Timucin, E., Venturini, A. et al. Understanding thermal and organic solvent stability of thermoalkalophilic lipases: insights from computational predictions and experiments. *J Mol Model* 26, 122 (2020). <https://doi.org/10.1007/s00894-020-04396-3>

Núria Prim, Marta Sánchez, Cristian Ruiz, F.I Javier Pastor, Pilar Diaz, Use of methylumbelliferyl-derivative substrates for lipase activity characterization, *Journal of Molecular Catalysis B: Enzymatic*, Volume 22, Issues 5–6, 2003, Pages 339-346, [https://doi.org/10.1016/S1381-1177\(03\)00048-1](https://doi.org/10.1016/S1381-1177(03)00048-1)

We have clarified the sentence as follows:

As a model reaction we selected the hydrolysis of the analytical substrate 4-Methyl Umbelliferone Butyrate (MUB) into the fluorescent product 4-Methyl Umbelliferone (MU)^{52,59} (Figure 1F).

7) The kinetics analysis (table 1) shows a very moderate increase in the Vmax of the heterogenous system (3.8 uM s⁻¹) compared to the free enzyme (2.1 uM s⁻¹). Can the authors explain why the threefold enhancement effect observed for the initial rate of the heterogenous system vs. the homogenous system is not observed for Vmax?

Answer from the authors: The initial rate of an enzyme which follows Michaelis Menten kinetics is a function of both the v_{max} and K_m. As the K_m is also influenced by condensation the change in the overall initial rate is a combination of changes in the v_{max} and the K_m. Prompted by this comment from the reviewer we carefully revisited the two phase kinetic model analysis, re-performing fitting and updating estimated values, which confirms that indeed there is both an increase in the k_{cat} and a decrease in the K_m of the enzyme in the dense phase.

8) I am not sure that the analysis with 750 mM NaCl demonstrate that there is no enhancing effect of the LAF1 fusion, as at this high ionic strength the enzyme activity is probably hindered.

Answer from the authors: Indeed, the increase in ionic strength impacts the enzymatic activity of BTL2. As can be seen in Figure 1H, the initial rate of BTL2 is decreased slightly when salt is added to the buffer. However, for this analysis the important aspect is that there is no difference in initial rate between BTL2 and Laf1-BTL2-Laf1, thus showcasing that the activity of the enzyme is independent of the conjugation with the LCD. This conclusion is further supported by the analysis at low enzyme concentrations below the c_{sat} (20nM) (Figure 1H), in the absence of phase separation.

9) The analysis confirming that the LAF1 condensates alone are not catalytic (Fig. S5) is important and I think that the authors wisely included it.

Answer from the authors: We thank the reviewer for this positive feedback.

10) How did the authors overcome potential changes in the emission intensity of the fluorescent product in the heterogenous system due to scattering? it is worthwhile mentioning it in the text.

Answer from the authors: We thank the reviewer for highlighting this important control experiment. No differences in fluorescence of the MU product could be observed in the presence or absence of condensates. The following sentence was added in the Results section to clarify, and the corresponding figure was added to the supplementary information.

We verified that the dense phase does not interfere with the fluorescence signal of MUB by measuring the fluorescence of the molecule in the absence and presence of Laf1-BTL2-Laf1 condensates. No significant difference between the two conditions was observed (Supplementary Figure 5).

Supplementary Figure 5: MU Fluorescence at 450 nm in 24 mM Tris buffer, 10 mM NaCl, pH 7.5 in the presence (red) and absence (black) of Laf1-BTL2-Laf1 condensates (total protein concentration: 0.5 μ M) Samples were measured at least in duplicate. Error bars indicate the standard error of the mean.

11) Page 7 kinetic model: the authors state that the evaluation of the activity coefficient in the dense phase requires assumption about the diffusion of the substrate. Since MUB is a fluorescent molecule, can the authors analyze its diffusion in the dense phase by FRAP?

Answer from the authors: We thank the reviewer for pointing this out. Indeed, in light also of previous work done in our lab (Gil-Garcia *et. al.*) diffusion limitations within the condensate can occur even with small molecules. We measured the diffusion coefficient of ATTO-565 within Laf1-BTL2-Laf1 condensates by fluorescence correlation spectroscopy (FCS) (the product, MU, could not be used due to laser settings of the instrument). Indeed, the measured value indicates that the time scale of diffusion and reaction are comparable. This indicates that the kinetic parameters evaluated for the dense phase by the model represent the lower bounds. This explanation was included in the main text:

We note that our model is built on the assumptions of instantaneous phase equilibrium and absence of diffusion limitations. However, using fluorescence correlation spectroscopy (FCS) we determined a diffusion coefficient of $1.015\mu\text{m}^2\text{s}^{-1}$, which if a condensate diameter of $1\mu\text{m}$ is considered, corresponds to a characteristic time of diffusion (τ_{diff}) of small molecules within condensates of 985 ms (see Materials and Methods and Supplementary Figure 8), which is in the same order of magnitude as the characteristic time of reaction ($t_{\text{react}}=1/k_{\text{cat}}= \text{ms}$). Therefore, the fitted kinetic parameters evaluated in the dense phase with the simplified model are an underestimation of the k_{cat} and an overestimation of the K_M .

12) Do the authors know experimentally what is the partitioning coefficient of the product?

Answer from the authors: The partitioning coefficient of an analogue molecule (Resorufin) was measured using confocal microscopy (see Suppl. Fig. S15) and was determined to be ca. 6.3. This proxy molecule is structurally very similar to the true MU product (see image below). Due to laser settings in the available confocal microscope MU could not be directly visualized.

13) The authors state that the kinetics analysis supports their initial hypothesis about conformational changes of the enzyme in the dense phase. While I agree that this is very likely to happen, there is no direct experimental evidence

that reaffirms this hypothesis. I suggest that the authors should either re-phrase this statement or perform additional biophysical experiments to directly analyze the conformation of the enzyme in different viscosities/conditions that can mimic the dense phase.

Answer from the authors: We appreciate the comment of the reviewer and we have rephrased the sentence accordingly:

It is very likely that this increase in enzymatic activity is due to a conformational change of the enzyme within the condensates (Figure 1A), driven by the more apolar environment and the local increase in enzyme concentration.

Minor points:

The figures include a lot of white spaces. Although not critical, I would recommend re-organizing the panels in Figure 1 to avoid these spacings.

Answer from the authors: We thank the reviewer for pointing this out. Figure 1 has been adapted to minimize white space.

Figure 1: Microreactor design and characterization. **A)** Schematic illustration of the conformational change of BTL2 (red and grey regions denote the open and closed conformations respectively). PDB open: 2w22, PDB closed: 1j13 (analogue from *G. Stearothermophilus*). **B)** Schematic representation of the design of the fusion protein forming condensates. **C)** Representative brightfield and fluorescence confocal images of Laf1-BTL2-Laf1 condensates stained with 25 μM SNARF-1 (0.5 μM protein, 24 mM Tris Buffer at pH 7.5, 10 mM NaCl). **D)** Phase diagram of Laf1-BTL2-Laf1 measured at 0.5 μM protein concentration at different pH and salt conditions. Blue circles and red crosses denote presence and absence of phase separation respectively. All buffers contain Tris/BisTris Buffers with 10mM ionic strength **E)** PRODAN assay to evaluate the apparent polarity of the Laf1-BTL2-Laf1 condensate environment (9 replicates). Fluorescence emission maximum λ_{max} measured at 0.5 μM Laf1-BTL2-Laf1 in 24 mM Tris at pH 7.5 and 10 mM NaCl, 100 μM PRODAN. For comparison, values measured in water (3 replicates) and isopropanol (6 replicates) are included. **F)** Initial rates of MUB hydrolysis catalyzed by BTL2 in the absence and presence of 10% isopropanol (0.1 μM BTL2, 0.25mM MUB, 24mM Tris, pH 7.5). **G)** Representative kinetic curves of the MUB hydrolysis reaction catalyzed by BTL2 (black) and Laf1-BTL2-Laf1 (red) (0.5 μM protein, 0.25 mM MUB, 10 mM NaCl, 24 mM Tris, pH 7.5, 5 replicates each). Error bars denote the standard error of the mean. **H)** Initial rates of MUB hydrolysis by BTL2 (black) and Laf1-BTL2-Laf1 (red) in 24 mM Tris, pH 7.5 at low ionic strength (10 mM NaCl, and 0.5 μM enzyme, 5 replicates), high ionic strength (750 mM NaCl and 0.5 μM enzyme, 3 replicates) and at an enzyme concentration below c_{sat} (10 mM NaCl and 0.02 μM enzyme, 4 replicates). All experiments were performed with 0.25 mM MUB. **I)** Enzymatic rate of BTL2 (black) and Laf1-BTL2-Laf1 (red) measured at increasing substrate concentrations (0.5 μM enzyme, 24 mM Tris, 10 mM NaCl, pH 7.5, at least 4 replicates). Continuous lines represent fits according to a Michaelis-Menten model. Error bars represent the standard error of the estimate.

Panels g-i): created in BioRender. Stoffel, F. (2025) <https://BioRender.com/sfpppex>, <https://BioRender.com/9ecyoX6>;

panel a): created in BioRender. Stoffel, F. (2025) <https://BioRender.com/bb8rbcq>;

panel b): created in BioRender. Stoffel, F. (2025) <https://BioRender.com/r5e105g>